# Multi-omics profiles of chronic low back pain and fibromyalgia—Study protocol

**Michele Curatolo**[1,2]*, **Abby P. Chiu**[1,2], **Catherine Chia**[1,2], **Ava Ward**[1], **Savera Khan**[1], **Sandra K. Johnston**[2,3], **Rebecca M. Klein**[4], **Darrell A. Henze**[4], **Wentao Zhu**[1], **Daniel Raftery**[1]

1 Department of Anesthesiology and Pain Medicine, University of Washington, Seattle, Washington, United States of America, 2 CLEAR Center for Musculoskeletal Research, University of Washington, Seattle, Washington, United States of America, 3 Department of Radiology, University of Washington, Seattle, Washington, United States of America, 4 Department of Neuroscience, Merck & Co., Inc., Rahway, New Jersey, United States of America

* curatolo@uw.edu

## Abstract

### Background

Chronic low back pain (CLBP) and fibromyalgia (FM) are leading causes of suffering, disability, and social costs. Current pharmacological treatments do not target molecular mechanisms driving CLBP and FM, and no validated biomarkers are available, hampering the development of effective therapeutics. Omics research has the potential to substantially advance our ability to develop mechanism-specific therapeutics by identifying pathways involved in the pathophysiology of CLBP and FM, and facilitate the development of diagnostic, predictive, and prognostic biomarkers. We will conduct a blood and urine multi-omics study in comprehensively phenotyped and clinically characterized patients with CLBP and FM. Our aims are to identify molecular pathways potentially involved in the pathophysiology of CLBP and FM that would shift the focus of research to the development of target-specific therapeutics, and identify candidate diagnostic, predictive, and prognostic biomarkers.

### Methods

We are conducting a prospective cohort study of adults ≥18 years of age with CLBP (n=100) and FM (n=100), and pain-free controls (n=200). Phenotyping measures include demographics, medication use, pain-related clinical characteristics, physical function, neuropathic components (quantitative sensory tests and DN4 questionnaire), pain facilitation (temporal summation), and psychosocial function as moderator. Blood and urine samples are collected to analyze metabolomics, lipidomics and proteomics. We will integrate the overall omics data to identify common mechanisms and pathways, and associate multi-omics profiles to pain-related clinical characteristics, physical function, indicators of neuropathic pain, and pain facilitation, with psychosocial variables as moderators.

**Data availability statement:** No datasets were generated or analyzed during the current study. All relevant data from this study will be made available upon study completion and publications.

**Funding:** This research was 1) Supported by grant funds from Merck Sharp & Dohme LLC, a subsidiary of Merck & Co., Inc., Rahway, NJ, USA. Grant recipients: M.C. and D.R.. Grant number not applicable. URL: www.merck.com. The sponsor did not play any role in the study design, data collection and analysis, decision to publish, or preparation of the manuscript. However, scientists of the sponsor (R.M.K. and D.A.H.) are part of the research team, and contributed to the study design, will contribute to the data analysis, decision to publish, and preparation of the manuscript. 2) Supported in part by the University of Washington Clinical Learning, Evidence, And Research (CLEAR) Center for Musculoskeletal Disorders, Administrative, Methodologic Cores - CLEAR is supported by NIAMS/NIH. Grant recipients: M.C. and S.K.J. Grant number P30AR072572. URL: https://theclearcenter.org. The sponsor did not play any role in the study design, data collection and analysis, decision to publish, or preparation of the manuscript.

**Competing interests:** The authors have declared that no competing interests exist.

## Discussion

Our study addresses the need for a better understanding of the molecular mechanisms underlying chronic low back pain and fibromyalgia. Using a multi-omics approach, we hope to identify converging evidence for potential targets of future therapeutic developments, as well as promising candidate biomarkers for further investigation by biomarker validation studies. We believe that accurate patient phenotyping will be essential for the discovery process, as both conditions are characterized by high heterogeneity and complexity, likely rendering molecular mechanisms phenotype specific.

## Introduction

Chronic pain affects 20% of the population and represents a leading cause of suffering, disability, and socioeconomic burden [1,2]. Chronic low back pain (CLBP), defined as low back pain (LBP) lasting 3 months or longer [3,4], is the leading determinant of disability worldwide [5]. Fibromyalgia (FM) is a chronic pain condition characterized by widespread pain that is not explained by tissue lesions. The prevalence of chronic widespread pain ranges across studies, with most estimates between 10% and 15% [6,7].

Therapies commonly used for CLBP and FM include physical therapy, various medications, psychological therapies, and a wide assortment of complementary and alternative therapies, all of which show little to modest effectiveness [8,9]. Nearly 50% of patients with high-impact CLBP and 24% of patients with FM use prescription opioids [10,11]. The very high rate of opioid use in prevalent conditions makes CLBP and FM major contributors to the opioid epidemic.

Current pharmacological treatments do not target molecular mechanisms driving CLBP and FM. Because we know very little about molecular mechanisms promoting CLBP and FM, we are unable to target them, and there is very little mechanistic evidence for the use of any currently available therapeutics. Furthermore, no validated biomarkers that allow a mechanism-based diagnosis, prediction of treatment effects, and prediction of the course of the pain condition are available.

Omics research has the potential to substantially advance our ability to develop mechanism-specific therapeutics by identifying pathways involved in the pathophysiology of CLBP and FM, and facilitate the development of diagnostic, predictive, and prognostic biomarkers. In a review, we identified only 19 studies that have applied a metabolomics approach to chronic pain [12]. Most studies were characterized by small sample sizes, heterogeneity of the patient population and limited phenotyping. Nevertheless, some consistency has been found for alterations in glutamate and testosterone metabolism, and metabolic imbalances caused by the gut microbiome, but further and well-powered studies are needed. Inflammatory markers may be associated with CLBP, but a systematic review concluded that high quality longitudinal studies are needed to confirm these associations [13]. Similarly, inflammatory mediators have been studied for FM, with inconsistent results [14,15]. Plasma proteomics and lipidomics have received so far limited attention in CLBP [16] and FM [17].

To date, omics studies have been sparse and therefore provided insufficient information on the molecular pathways involved in CLBP and FM. One limitation that has been identified in past studies is the focus on single omics, as opposed to a multi-omics approach [16]. In addition, most studies have been characterized by limited patient phenotyping. Because of the complexity of CLBP and FM, it is likely that molecular mechanisms are specific to sub-populations that can be identified by careful phenotyping, rather than to CLBP and FM collectively.

We will conduct a blood and urine multi-omics study in comprehensively phenotyped and clinically characterized patients with CLBP and FM. Our aims are to identify molecular pathways potentially involved in the pathophysiology of CLBP and FM that would shift the focus of research to the development of target-specific therapeutics, and identify candidate diagnostic, predictive, and prognostic biomarkers. Because of the very limited previous research, we do not have an *a priori* hypothesis involving specific molecular pathways, and will perform unbiased analyses to allow discovery of potentially relevant molecular pathways and candidate biomarkers.

## Methods

### Study design

We are conducting a prospective cohort study of adults ≥18 years of age with CLBP and FM, and pain-free controls. We recruit CLBP and FM participants from the University of Washington (UW) Center of Pain Relief, physical therapy and sports medicine clinics. Additionally, both chronic pain and pain-free controls are recruited through UW's Institute of Translational Health Sciences (ITHS) community research recruitment site. Study participation is a single visit.

Ethics approval for this study was provided by the Institutional Review Boards at the University of Washington (#STUDY00017121). This study qualified as minimal risk, and the Institutional Review Board waived the requirement for documentation of consent. Consequently, all participants provide verbal or web-based informed consent for participation and all data collection prior to being enrolled. The project has started on August 1, 2023, and is expected to be completed by July 31, 2026.

### Participants and eligibility criteria

We will plan to enroll 100 participants with CLBP, 100 with FM, and 200 pain-free controls. The inclusion and exclusion criteria are presented in Table 1.
Pain duration for those with CLBP will be at least 3 months duration of low back pain according to the definition by the International Association for the Study of Pain (IASP) [3,4]. FM participants are included based on the revised criteria of the American College of Rheumatology, when all of the following criteria are met: pain in at least 4 of 5 regions, symptoms are present at a similar level for at least 3 months, widespread pain index (WPI) ≥7 and symptom severity score (SSS) score ≥5 or WPI of 4–6 and SSS score ≥9 [18]. Pain-free controls are matched as close as possible on sex, age, and body mass index. We exclude anyone with current active cancer, undergoing cancer treatment, diagnosed with non-skin cancer within the last 2 years or who are pregnant by self-report. Additionally, those in the pain-free control group are excluded for any continuous pain lasting greater than 1 day in the last 2 weeks and pain at any time of the study procedures.

### Recruitment

We are using automated data generated weekly from the electronic health record (EHR) at the UW Center of Pain Relief, physical therapy and sports medicine clinics through the technical center Perioperative & Pain initiatives in Quality Safety Outcome (PPiQSO). The report uses the M54.4* and M54.5*International Classification of Diseases, Tenth Revision, Clinical Modification (ICD10) diagnosis codes for CLBP and FM. The research coordinators review the EHR for inclusion and exclusion criteria to generate a viable list of potential participants. Additionally, people from the community can self-recruit through the UW's ITHS research recruitment site or by responding to recruitment flyers displayed at UW sites. Inclusion and

exclusion criteria as well as a QR code and contact information are included in the ITHS and flyer recruitment materials. Potential participants receive a call from the research coordinators to discuss the study and confirm their eligibility. Those who are interested are provided links for self-enrollment including a study consent and surveys, and are scheduled for the in-person research study visit. The study visit includes blood and urine collections, a comprehensive medical history and quantitative sensory testing (QST).

## Phenotyping measures

Data is collected with study participants by telephone, on-line surveys and a single in-person study visit. The measures are presented in Table 2.

From the EHR, we extract sex, age, body mass index (BMI), and all pain-related diagnoses. Using participant surveys, we record race, ethnicity, alcohol usage, smoking status [19], duration of FM and CLBP, and use during the past 30 days of opioids (with doses), cannabinoids, antidepressants, and anticonvulsants for pain.

We selected the phenotyping measures based on past research, our previous work, and conceptual understandings of pain mechanisms. We have selected the shortest possible validated measures to minimize participant burden and increase feasibility of obtaining complete data. We obtain phenotyping data in the domains outlined below. Measures that do not apply to the pain-free control participants are marked*: 1) Pain-related clinical characteristics (pain intensity* [20], Widespread Pain Index* and Symptom Severity Score* [18], and pressure pain sensitivity [21]; 2) physical function [22]; 3) neuropathic components (quantitative sensory tests [23], and the DN4 questionnaire* [24]); 4) pain facilitation (temporal summation) [25]; and 5) psychosocial function as moderator (depression [26] anxiety [27] and catastrophizing [28]).

**Pain-related clinical characteristics.** Pain intensity is rated using the 0–10 numerical rating scale (NRS) that is part of the PROMIS-29 Profile v2.0 [20]. Patients are asked to rate the intensity of their LBP or FM pain on average in the past 7 days, and at the time of study visit (0 = no pain, 10 = worst imaginable pain). The Patient-Reported Outcomes Measurement Information System (PROMIS) is a standardized patient-reported outcome assessment of

**Table 1. Inclusion and exclusion criteria.**

|  | Chronic low back pain | Fibromyalgia | Pain-free controls |
|---|---|---|---|
| Inclusion | At least 3 months duration of low back pain | Pain in at least 4 of 5 regions<br>Symptoms have been present at a similar level for at least 3 months<br>Widespread pain index (WPI) ≥ 7 and symptom severity score (SSS) ≥ 5, or WPI of 4–6 and SSS score ≥ 9<br>The diagnosis is valid irrespective of other diagnoses |  |
| Exclusion | <18 years old<br>Unable to provide informed consent to participate in the study or provide accurate data<br>Unable to speak or write in the English language<br>Currently have cancers or are undergoing treatment for cancer (except for skin cancers)<br>Diagnosed with any non-skin cancer within the last 2 years<br>Pregnant women by self-report | <18 years old<br>Unable to provide informed consent to participate in the study or provide accurate data<br>Unable to speak or write in the English language<br>Currently have cancers or are undergoing treatment for cancer (except for skin cancers)<br>Diagnosed with any non-skin cancer within the last 2 years<br>Pregnant women by self-report | <18 years old<br>Unable to provide informed consent to participate in the study or provide accurate data<br>Unable to speak or write in the English language<br>Currently have cancers or are undergoing treatment for cancer (except for skin cancers)<br>Diagnosed with any non-skin cancer within the last 2 years<br>Pregnant women by self-report<br>Any continuous pain lasting > 1 day during the last two weeks<br>Any pain at the time of the study procedures. |

**Table 2. Measures in the chronic low back pain and fibromyalgia groups.**

| Construct | Measures |
|---|---|
| Sex at birth | Sex at birth: (male, female, intersex, unknown, prefer not say), retrieved from EHR |
| Date of birth | Date of birth: (mm/dd/yyyy), retrieved from EHR |
| Body-mass index | Retrieved from EHR |
| All pain-related diagnosis | Retrieved from EHR |
| Race | Race (Choose all that apply): American Indian or Alaska Native, Asian, Black or African American, Native Hawaiian or Pacific Islander, White, Unknown |
| Ethnicity | Ethnicity: Hispanic or Latino, not Hispanic or Latino, unknown |
| Smoking status | Do you smoke or have you smoked?<br>I currently smoke<br>I have previously smoked<br>I have never smoked |
| Alcohol use | Over the past 3 months, how often do you typically drink alcohol? One serving is about 1 small glass of wine (5 oz), 1 beer (12 oz), or 1 single shot of liquor.<br>I do not drink alcohol<br>Less than 3 servings per week<br>½ to 1 serving per day<br>1 to 3 servings per day<br>More than 3 servings per day |
| Medication use | Have you been using prescribed medication [opioid], [oral or inhaled cannabinoids], [anti-depressants], [anti-convulsant] during the past 30 days?<br>Yes<br>No<br>Not sure<br>If yes, choose the type of [opioid], [oral or inhaled cannabinoids], [anti-depressants], [anti-convulsant] and include the dosage and frequency |
| Pain duration | How long have you had the type of pain for which you are enrolled in this study? Please list the number of months or years.<br>How long has this type of pain been an ongoing problem for you?<br><3 months<br>3-6 months<br>>6 months - <1year<br>1-5 years<br>>5 years |
| Pain intensity, average of past 7 days | 0-10 NRS from PROMIS-29 Profile v2.0. |
| Pain intensity, at the time of QST testing | 0-10 NRS from PROMIS-29 Profile v2.0. completed during QST |
| Widespread Pain Index | Clinical Fibromyalgia Diagnostic Criteria – Part 1. Widespread Pain Inventory. |
| Symptom Severity Score | Clinical Fibromyalgia Diagnostic Criteria – Part 1. Symptom Severity Score – Part 2a, 2b. |
| Physical function | PROMIS Short Form v2.0 – Physical Function 4a |
| Fatigue | PROMIS Short Form v1.0 – Fatigue 4a |
| Sleep | PROMIS Short Form v1.0 – Sleep Disturbance 4a |
| Neuropathic pain | DN4 questionnaire, two items completed by the patients and the other two by the QST examiner |
| Depression | PHQ-2 |
| Generalized anxiety disorder | GAD-2 |
| Pain Catastrophizing | PCS-4 |

EHR: electronic health records. QST: quantitative sensory testing. DN4: Douleur Neuropathique-4 item. PHQ-2: Patient Health Questionnaire-2 items. GAD-2: Generalized Anxiety Disorder- 2 items. PCS-6: Pain Catastrophizing Scale-6 items.

health-related quality of life [29]. The pain intensity rating is responsive to clinical change among adults with chronic musculoskeletal pain [30]. Patients and content experts were extensively involved in its development [31].

The Widespread Pain Index (WPI) and Symptom Severity Score (SSS) are used for the diagnosis of FM and the quantification of FM symptoms [18]. The questionnaires are administered also to the CLBP cohort to additionally determine pain locations and potential FM symptoms in patients who do not meet the threshold for the diagnosis of FM. Clinically, patients with pain in a single location differ from those with pain in multiple bodily locations [32], and may display different omics profiles.

Pressure pain sensitivity is a clinically relevant characteristic, and its assessment is part of the routine clinical examination of CLBP and FM. Pressure pain sensitivity is quantified by pressure pain thresholds (PPT). We have previously shown that PPTs are lowered in LBP and FM; they rank first among 26 quantitative sensory tests in discriminating patients with acute and chronic LBP from healthy controls [33–36]. We apply a widely used protocol [37], using an analog pressure algometer (Wagner FDK 40) with a surface area of 1 cm². The pressure is increased at a rate of 0.5 kg-force/s, using 1 Hz metronome ticks as a guide. Pain threshold is defined as the point at which the pressure sensation turns to pain. If 10 kg-force is reached before pain is reported, this value is considered as threshold. The test is performed at the site of most severe pain in the two patient groups, and on the suprascapular region, right or left (randomly selected) in the control group [35,36]. Testing the control group will allow the determination of pressure hypersensitivity in patients.

**Physical function.** Physical function is assessed by the PROMIS Physical Function 4-item short form. This measure has favorable psychometric properties, including in patients with CLBP [29,38], has been recommended by the NIH Task Force on Research Standards for Chronic Low Back Pain [22], has construct validity, and is responsive to clinical change among adults with chronic musculoskeletal pain [30]. The VA Work Group on Core Outcome Measures for Chronic Musculoskeletal Pain Research recommended this scale as a core secondary outcome measure [39].

**Neuropathic components.** Features of neuropathic pain have been detected in subsets of patients with CLBP using validated questionnaires and quantitative sensory testing (QST) [40–42]. Neuropathic components of CLBP are still under-recognized and under-treated [43]. Small fiber neuropathy and nociceptor abnormalities have been detected in FM patients [44,45]. To assess neuropathic components, we will use quantitative sensory testing (QST) and the DN4 questionnaire.

We apply a selection of bedside QST according to published guidelines [23], a standardized and widely used protocol [46], and our previous work with QST in LBP and FM [33–36,47–59]. The QST are performed in CLBP and FM groups at the site of most severe pain, and in all three groups on a non-painful site of the upper extremity, right or left (randomly selected) [35,36]. Testing the control group will allow the determination of sensory abnormalities in patients, specifically reduced sensation with different stimulation modalities and pain to innocuous stimuli (allodynia), as recommended by diagnostic guidelines for neuropathic pain [23,46,60,61]. A) Light brushing: soft brush, applied for 1s. B) Light pressure: 5.07/10 gm monofilament, applied so that it bends and maintained during bending for 1s [62]. C) Coolness: 128Hz tuning fork that has been in ice water (2–5 °C) ≥30 min, applied for 5s. Patients will rate the sensation of brushing, pressure, and coolness from 0 (felt nothing) to 10 (extremely intense feeling). In addition, patients are asked if each of the three modalities causes pain, a positive answer indicating dynamic, static, and cold allodynia, respectively.

In addition, we administer the DN4 questionnaire, which has been validated for neuropathic pain and has detected neuropathic components in CLBP and FM [24,63,64].

**Pain facilitation.** Temporal summation refers to an increased perception of pain in response to repeated stimuli of equal intensity, indicating facilitation of pain modulation processes [65–67]. Temporal summation is enhanced in CLBP and FM [35,36,48,49]. We use a pinprick stimulator at 256 mN. Patients first rate the intensity of a single stimulus (0 = no pain, 100 = worst pain imaginable). We then apply a series of 10 identical stimuli with a frequency of 1 Hz using 2 Hz metronome ticks as a guide, with patients rating the pain intensity at the end of the series. This sequence is applied at the same three sites as the QST. Pain facilitation is assessed as the difference between pain rating after 10 stimuli and pain rating after 1 stimulus: the higher the difference, the higher the facilitation of pain modulation processes [25].

**Moderator: psychosocial function.** Psychosocial factors play important roles in the development and maintenance of chronic pain, and are associated with alterations in central pain modulating processes [68,69]. Pain catastrophizing is associated with persistent pain and disability [53,70,71], and with changes in brain areas involved in pain processing [72] after lumbar spine surgery. Depression increases the risk of developing chronic pain, and depression and anxiety are risk factors for worse chronic pain outcomes [73,74].

We assess depressive and anxiety symptoms using the 2-item PHQ-2 [26] and the 2-item GAD-2 [27], respectively, which comprise the PHQ-4. These are valid, reliable measures [75]. The VA Work Group on Core Outcome Measures for Chronic Musculoskeletal Pain Research recommended these scales as core secondary outcome measures [39]. We assess catastrophizing using the 4-item Pain Catastrophizing Scale, which is highly correlated with the original full Pain Catastrophizing Scale and has satisfactory internal consistency [28].

## Blood and urine samples

Blood samples (6 mL) are collected from subjects in the morning (if possible) to reduce variability due to diet and circadian rhythm. Fasting is preferred but not required, and fasting status is recorded. Collection times are noted so that it can be used as a covariate or modeled. Samples are collected in Lithium Heparin tubes and placed on ice immediately and then transported to the lab for processing and frozen at -80 °C until analyzed. Samples are processed by removing the red blood cells and buffy coat and collecting up to 10 plasma aliquots of 0.5 ml each.

Midstream urine samples (at least 10 mL) are collected and similarly placed on ice, then transported to the lab, processed and frozen at -80 °C until analyzed. Prior to freezing, the urine samples are split into at least 10–1 ml aliquots.

## Data analysis

Plasma and urine metabolomics and lipidomics analysis will include global and targeted metabolite profiling of aqueous metabolites, lipids, and oxylipin signaling lipids. Plasma proteomics will include the Olink Explore HT panel which covers proteins across a broad range of biological pathways including cytokines. We will start with metabolomics analyses and will then integrate the data with the proteomics and lipidomics analysis.

We will compare blood and urine in the three groups and associate multi-omic profiles with the above phenotyping characteristics. After data pretreatment to correct instrument drift and impute missing values, biomarker discovery will be performed using two-sided non-parametric tests with a p-value = 0.05. We will set the false discovery rate at 0.1 in adjusting for multiple testing, to account for potential false positives. In addition, we will screen potential biomarkers using area under the receiver operating curve (AUROC) and effect size, which are important performance parameters for biomarker development. Grouping the

metabolites and proteins by class or pathway is an approach we will evaluate to not unduly limit the biomarker discovery effort. Validation of existing biomarkers reported in the literature will be made using our targeted platform, a Sciex 6500 LC-MS/MS system, and non-parametric one-sided tests. In addition to simply analyzing the individual metabolites, pathway analyses will allow us to identify signals in a more statistically powerful way. For example, we can perform pathway analysis using the pathway tool provided by MetaboAnalyst ([https://www.metaboanalyst.ca/](https://www.metaboanalyst.ca/)), which characterizes pathways both by statistically significance and by the number of pathway metabolites that differ between pain patients and controls. Significantly altered pathways associated with FM and CLBP will provide key information towards identifying potential therapeutic targets.

We will also use multivariate statistical approaches to develop metabolic profiles of pain using, for example, partial least squares discriminant analysis (PLS-DA), as well as hierarchical clustering approaches to identify sets of metabolites that can be used to characterize the metabolic signatures of either widespread or localized pain. Patient sex and measures of psychosocial function will be analyzed as moderators: depression (PHQ-2), anxiety (GAD-2), and catastrophizing (4-item Pain Catastrophizing Scale).

We will integrate the metabolomics with the overall omics data to identify common mechanisms and pathways. We will associate multi-omics profiles to pain-related clinical characteristics, physical function, indicators of neuropathic pain, and pain facilitation. We anticipate identifying omics profiles and pathways that are associated with pain phenotypes. This will provide information on mechanisms of specific importance to CLBP and FM that are linked with specific clinical pain phenotypes. We will adopt a holistic approach to evaluate the association between omics profiles and phenotyping that will consider convergent associations with multiple pain-related phenotyping domains. This involves a primary analysis and exploratory analyses, minimizing the risk of false-negative results. Sensitivity analyses will be performed to study the potential influence of outliers, such as patients with unusual pain diagnoses.

In multinomial regression models, the dependent variables will be CLBP, FM, and pain-free (reference group) phenotypes. We will select the most important and significant omics using a penalized, LASSO regression model approach. In exploratory analyses, we will perform linear, logistic, and multinomial logistic regressions for continuous, binary, and categorical dependent variables, respectively, with the phenotyping measures as dependent variables. To minimize false-positive findings due to multiple testing, we will select the independent variables based on the concept of the false discovery rate [76], using the q-value (instead of p-value) as criterion for statistical significance [77,78]. The independent variables will be omics profiles that will display an association with the CLBP/FM vs. pain-free phenotype according to the primary analysis, with a Benjamini-Hochberg family wise error value < 0.1 (power = 0.8, correlations with Pearson's $R^2$ > 0.3, and sample size ≥ 50). This threshold allows for a greater tolerance of false positives. Sex and psychosocial variables will be included in the models as covariates to test whether associations between the metabolites and pain phenotypes are influenced by sex [79,80] and psychosocial variables. According to pain research guidelines, adjustment of the significance level for multiplicity is not required for exploratory analyses [81]. After our exploratory analyses are complete, we will map out the associations between omics profiles and phenotypes using regression analysis by point estimate and q-value.

To expedite the biomarker discovery process, we will initialize the statistical data workflow and preprocessing steps early in year 2, while the initial biomarker discovery will take place once we have acquired metabolomics data on 70 patients with low back pain, 70 patients with fibromyalgia and 140 controls, which we anticipate near the end of year 2. We will then treat the additional samples (30 per group) as an internal validation set, and combine both the discovery and validation sample sets for improved power.

**Sample size estimate.** Power calculations based on promising biomarker candidates from the literature indicate that we should be able to confirm the performance of previously reported pathways as well as discovery new omics profiles and putative biomarker candidates of similar or better performance [82]. Pain-free controls will be matched for sex, age and BMI as much as possible, without compromising the collection rate so as to not slow down the study. It is possible to adjust for these covariate factors and even to model their effects on metabolite levels as we have shown previously [83].

As shown in Table 3, assuming a sample size of 100 per group would allow us to validate all putative biomarkers, including metabolites with effect sizes as low as 0.4, as shown for lactate. Because we will be collecting 200 samples from healthy controls the power will be slightly better than in Table 3. For discovery, because we will be detecting multiple omics, each with many analytes, we will need to be cognizant of the potential for false discoveries. In addition to performing cross validation using training and testing sets of samples, we will also group omics

based on, for example, lipid or metabolite class or metabolic pathway, or biological pathway to reduce the number of variables and increase statistical power. See above for a description of the proposed statistical analysis. The large sample size for this study will help reduce the false discovery rate, but at the same time allow us to identify a relatively large number of significant omics that can be used to identify altered pathways, and hopefully, specific signaling pathways that could ultimately be targeted for drug development. We should also be able to identify key pathways and biomarkers within the female or male.

## Discussion

Metabolomics studies in chronic pain have been sparse, as we have shown in a previous review [12]. Most studies in chronic low back pain had a partial focus, such as on single metabolites or pathways [84], tissue metabolites [85], or association with radiological findings

**Table 3. Basis for the sample size calculation.**

| Metabolite | FM | Control | Sample | Method | Power for detecting metabolite | | | | |
|---|---|---|---|---|---|---|---|---|---|
| | | | | | Effect size | P-value | n=25 | n=50 | n=100 |
| 2-hydroxybutyrate | 19 | 19 | Urine | NMR | 0.72 | 0.0001 | 0.704 | 0.946 | 0.999 |
| Succinate | | | | | 0.61 | 0.0001 | 0.558 | 0.852 | 0.990 |
| Taurine | | | | | 0.52 | 0.0007 | 0.567 | 0.726 | 0.954 |
| Tyrosine | | | | | 0.45 | 0.0029 | 0.468 | 0.603 | 0.884 |
| Lactate | | | | | 0.42 | 0.0044 | 0.428 | 0.670 | 0.837 |
| Hypoxanthine | 22(F) | 22(F) | Serum | HPLC-UV | 9.78 | <0.00001 | 1.000 | 1.000 | 1.000 |
| Xanthine | | | | | 8.64 | <0.00001 | 1.000 | 1.000 | 1.000 |
| Inosine | | | | | 8.46 | <0.00001 | 1.000 | 1.000 | 1.000 |
| lysoPC(14:0) | 22 | 21 | Plasma | LC-MS | 1.10 | 0.000036 | 0.968 | 1.000 | 1.000 |
| PC(18:1(9Z)/0:0) | | | | | 0.74 | 0.00013 | 0.727 | 9.956 | 0.999 |
| lysoPC(22:6) | | | | | 0.72 | 0.0024 | 0.704 | 0.946 | 0.999 |
| lysoPC(18:2) | | | | | 0.64 | 0.0049 | 0.602 | 0.887 | 0.994 |
| Glutamate | 105 | 54 | Serum | LC-MS | 0.50[*] | 0.034 | 0.410 | 0.700 | 0.940 |
| Glutamine | | | | | 0.76[*] | 0.00035 | 0.766 | 0.970 | 1.000 |

[*]Estimated. FM: Fibromyalgia.

The metabolites are derived by our review on metabolomics in chronic pain: Teckchandani, S., Nagana Gowda, G.A., Raftery, D., & Curatolo, M. (2021). Metabolomics in chronic pain research. European Journal of Pain, 25, 313–326. 10.1002/ejp.1677.

[86]. Comprehensive studies on the metabolome are lacking, and the literature does not provide strong indications on which metabolic pathways are of relevance in chronic low back pain. The metabolic profile of fibromyalgia has been better studied. Some studies have focused on single metabolic pathways, such as purine [87]. Other ones adopted a comprehensive metabolomics approach, but the results have not converged to clearly identified pathways involved in the pathophysiology of fibromyalgia [88–91]. Despite the importance of lipid metabolism for the understanding of cellular function and alterations in processes related to disease, such as inflammation, lipidomics studies in low back pain and fibromyalgia are sparse and mostly limited to few target lipids [16]. Most of the proteomics literature has focused on immune and inflammatory markers [92]. A recent systematic review in fibromyalgia identified dysregulated proteins associated with oxidative stress response, but the correlations with pain outcomes was weak [17].

Our study addresses the need for a better understanding of the molecular mechanisms underlying chronic low back pain and fibromyalgia. Using a multi-omics approach, we hope to identify converging evidence for potential targets of future therapeutic developments, as well as promising candidate biomarkers for further investigation by biomarker validation studies. The results will inform future hypothesis-driven mechanistic studies to identify relevant targets for pharmacological development, as well as biomarker validation studies. We believe that accurate patient phenotyping will be essential for the discovery process, as both conditions are characterized by high heterogeneity and complexity, likely rendering molecular mechanisms phenotype specific.

## Acknowledgments

The authors thank the nurses and staff of the Center for Pain Relief for the logistic support and their contribution to sampling the specimens.

## Author contributions

**Conceptualization:** Michele Curatolo, Abby P. Chiu, Sandra K. Johnston, Rebecca M. Klein, Darrell A. Henze, Daniel Raftery.

**Data curation:** Abby P. Chiu, Catherine Chia, Ava Ward, Savera Khan, Sandra K. Johnston, Wentao Zhu.

**Formal analysis:** Savera Khan, Wentao Zhu, Daniel Raftery.

**Funding acquisition:** Michele Curatolo, Daniel Raftery.

**Investigation:** Michele Curatolo, Abby P. Chiu, Catherine Chia, Ava Ward, Savera Khan.

**Methodology:** Michele Curatolo, Abby P. Chiu, Ava Ward, Savera Khan, Sandra K. Johnston, Rebecca M. Klein, Darrell A. Henze, Daniel Raftery.

**Project administration:** Michele Curatolo, Savera Khan, Sandra K. Johnston.

**Resources:** Catherine Chia, Savera Khan.

**Software:** Savera Khan.

**Supervision:** Michele Curatolo, Daniel Raftery.

**Validation:** Savera Khan, Wentao Zhu.

**Writing – original draft:** Michele Curatolo.

**Writing – review & editing:** Michele Curatolo, Abby P. Chiu, Catherine Chia, Ava Ward, Savera Khan, Sandra K. Johnston, Rebecca M. Klein, Darrell A. Henze, Wentao Zhu, Daniel Raftery.

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
