## [Decision Letter · Decision Letter 0]

2 Jan 2025

PONE-D-24-43124Multi-Omics Profiles of Chronic Low Back Pain and Fibromyalgia – Study ProtocolPLOS ONE

Dear Dr. Curatolo,

Thank you for submitting your manuscript to PLOS ONE. After careful consideration, we feel that it has merit but does not fully meet PLOS ONE’s publication criteria as it currently stands. Therefore, we invite you to submit a revised version of the manuscript that addresses the points raised during the review process.een the reviews so that it's clear which advice the authors should follow

We look forward to receiving your revised manuscript.

Kind regards,

Claudia Sommer

Academic Editor

PLOS ONE

Journal Requirements:

Reviewers' comments:

Reviewer's Responses to Questions

**Comments to the Author**

1. Does the manuscript provide a valid rationale for the proposed study, with clearly identified and justified research questions?

Reviewer #1: Yes

2. Is the protocol technically sound and planned in a manner that will lead to a meaningful outcome and allow testing the stated hypotheses?

Reviewer #1: Yes

3. Is the methodology feasible and described in sufficient detail to allow the work to be replicable?

Reviewer #1: No

4. Have the authors described where all data underlying the findings will be made available when the study is complete?

Reviewer #1: No

5. Is the manuscript presented in an intelligible fashion and written in standard English?

Reviewer #1: Yes

6. Review Comments to the Author

You may also provide optional suggestions and comments to authors that they might find helpful in planning their study.

Reviewer #1: PLOS ONE

Manuscript Number: PONE-D-24-43124

Title: Multi-Omics Profiles of Chronic Low Back Pain and Fibromyalgia – Study Pro-tocol

Revision 1:

Dear Authors,

Thank you for the opportunity to review this study protocol. It is an interesting ap-proach to investigate the complex mechanisms leading to chronic low back pain (CLBP) and fibromyalgia (FM). Although the protocol is already thoroughly written, I have identified some issues that I would like you to consider in order to further im-prove the reproducibility of the methodology and to be able to critically reflect on the results.

General comment:

• Please provide an appropriate reporting guideline according to which you have compiled your study protocol and also provide the appropriate checklist with the revised manuscript

• I could not identify as to whether your study protocol was pre-registered in an appropriate registry or as to whether you planned to do so; if not, please con-sider to do so and provide information in the revised manuscript accordingly

Introduction:

• Please provide a paragraph detailing potential a-priori hypotheses you might have based on your previous research findings

Methods:

• -Table 1:

• Currently, potential participants are 'only' screened for cancer as a 'red flag' exclusion criterion. However, there may be additional red flags that need to be addressed, e.g. (osteoporotic) fractures, cauda equine syndrome, system-ic inflammation ect. Please provide additional information if or where the protocol intends to screen for additional red flags or what additional measures will ensure that the conditions are CLBP and/or FM in nature.

• Neuropathic components:

• People will be screened for potential neuropathic components by various measures; if people are shown to have neuropathic pain components, will they be treated as a subset of CLBP, as the underlying pain mechanisms are different from those currently discussed in the context of CLBP? There-fore, please provide additional information on whether people presenting with neuropathic pain components will be treated as a subgroup and sepa-rately from the CLBP group.

• control group neuropathic component testing:

o Please provide more detailed information on the definition of "senso-ry abnormalities" based on the QST results of the testing of this group.

o Please provide more detailed information on the determination of “pain facilitation” based on the temporal summation results of the test-ing of this group.

Data analysis:

• Against the aspects mentioned before, please explain as to whether the sub-group of patients demonstrating neuropathic pain components could also/could not present an additional dependent variable in the multinomial regression analysis

Discussion:

As mentioned by the authors and reflected in the Methods section, this study is exploratory in nature. Therefore, the results will provide associations between multi-omics profiles and pain-related clinical features; however, these associations require further investigation before they can serve as "converging evidence for potential targets for future therapeutic develop-ment". Please be more cautious in this conclusion and provide a more de-tailed description of how the results could inform future research.

7. PLOS authors have the option to publish the peer review history of their article (what does this mean? ). If published, this will include your full peer review and any attached files.

**Do you want your identity to be public for this peer review?** For information about this choice, including consent withdrawal, please see our Privacy Policy .

Reviewer #1: **Yes: ** Prof. Dr. Katja Ehrenbrusthoff

---

## [Decision Letter · Decision Letter 1]

31 Jan 2025

Multi-Omics Profiles of Chronic Low Back Pain and Fibromyalgia – Study Protocol

PONE-D-24-43124R1

Dear Dr. Curatolo,

We’re pleased to inform you that your manuscript has been judged scientifically suitable for publication and will be formally accepted for publication once it meets all outstanding technical requirements.

Kind regards,

Claudia Sommer

Academic Editor

PLOS ONE

Additional Editor Comments (optional):

Reviewers' comments:

Reviewer's Responses to Questions

**Comments to the Author**

1. Does the manuscript provide a valid rationale for the proposed study, with clearly identified and justified research questions?

Reviewer #1: Yes

2. Is the protocol technically sound and planned in a manner that will lead to a meaningful outcome and allow testing the stated hypotheses?

Reviewer #1: Yes

3. Is the methodology feasible and described in sufficient detail to allow the work to be replicable?

Reviewer #1: Yes

4. Have the authors described where all data underlying the findings will be made available when the study is complete?

Reviewer #1: Yes

5. Is the manuscript presented in an intelligible fashion and written in standard English?

Reviewer #1: Yes

6. Review Comments to the Author

You may also provide optional suggestions and comments to authors that they might find helpful in planning their study.

Reviewer #1: Dear Authors,

Thank you for the careful revision of your manuscript along with a detailed point-by-point response to my comments. I see that all issues have been care-fully addressed and now consider the manuscript ready for publication.

7. PLOS authors have the option to publish the peer review history of their article (what does this mean? ). If published, this will include your full peer review and any attached files.

**Do you want your identity to be public for this peer review?** For information about this choice, including consent withdrawal, please see our Privacy Policy .

Reviewer #1: **Yes: ** Prof. Dr. Katja Ehrenbrusthoff

---

## [Editor Report · Acceptance letter]

PONE-D-24-43124R1

PLOS ONE

Dear Dr. Curatolo,

I'm pleased to inform you that your manuscript has been deemed suitable for publication in PLOS ONE. Congratulations! Your manuscript is now being handed over to our production team.

Kind regards,

on behalf of

Prof. Dr. Claudia Sommer

%CORR_ED_EDITOR_ROLE%

PLOS ONE